# Recruiters' perspectives of recruiting women during pregnancy and childbirth to clinical trials: A qualitative evidence synthesis

Vivienne Hanrahan[1]*, Katie Gillies[2], Linda Biesty[1]

**1** School of Nursing and Midwifery, National University of Ireland Galway, Galway, Ireland, **2** Health Services Research Unit, University of Aberdeen, Aberdeen, Scotland, United Kingdom

* v.hanrahan1@nuigalway.ie

**Data Availability Statement:** All relevant data are within the manuscript and its Supporting Information files.

## Abstract

### Introduction

Research on research is key to enhancing efficacy in trial methodology. Clinical trials involving women during pregnancy and childbirth are limited, with a paucity of data guiding evidence-based practice. Following a prioritisation exercise that highlighted the top-ten unanswered recruitment questions, this qualitative evidence synthesis was designed specifically to focus on the barriers and enablers for clinicians/healthcare professionals in helping conduct randomised trials within the context of recruitment during pregnancy and childbirth.

### Methods

The synthesis was undertaken using Thomas and Harden's three stage thematic synthesis method and reported following the ENTREQ guidelines. Using a pre-determined SPIDER strategy, we conducted a comprehensive search of databases; Pubmed, CINAHL, PsycINFO, EMBASE, and grey searches for records until January 2019. We included all reports of qualitative data on recruiter's experiences, perceptions, views of recruiting women during pregnancy and childbirth to clinical trials. Altogether 13,401 records were screened, resulting in 31 full-text reviews, of which five were eligible for inclusion. Quality was appraised using CASP. Data were extracted onto a specifically defined form. We used thematic synthesis to identify descriptive and analytical themes, and to interpret and generate theory. Confidence was assessed using GRADE-CERQual. The review protocol is publicly available (OSF https://osf.io/g4dt9/).

### Results

Five papers (representing four individual studies) from two different countries were included. All studies focused on the experiences of trial recruiters in the maternity setting. We identified four analytical themes; *Recruitment through a clinician's lens*, *Recruiters judgement on acceptability*, *From protocol to recruiters lived experience*, *Framing recruitment in context*. These were linked by an overarching theme combining beliefs and power.

**Funding:** This paper forms part of doctorial research project supported by the Health Research Board Trial Methodology Research Network (of Ireland) PhD scholarship awarded to VH. https://www.hrb-tmrn.ie/. The manuscript represents the views of the named authors only. The funders had no role in study design, data collection and analysis, decision to publish, or preparation of the manuscript.

**Competing interests:** The authors have declared that no competing interests exist.

## Conclusion

The overarching theme combining beliefs and power links the experiences and perceptions of recruiters. This synthesis shows a gap between the trial design study protocol and the recruiter's lived experience. Strategies such as collaborative trial design, mitigating gate-keeping behaviours, and training may support recruiters in their endeavour.

## Introduction

Recruitment of participants to randomised trials is essential to their successful completion yet difficulties in recruiting remains a significant challenge to trialists [1]. Trials frequently fail to reach recruitment targets [2], more recently a UK study [3] found just 50% of trials achieve optimal participant numbers. Consequently, these difficulties produce underpowered results of reduced clinical meaning and precision, which negatively impacts the quality and generalisability of the study [4]. Poor recruitment is also associated with increased costs, both in terms of time and money, and contributing to 'research waste' [5]. Furthermore, failure to reach a recruitment target can pose an ethical issue, in that a participant is exposed to an intervention which then proves to be of indeterminate benefit [4].

In acknowledging the inefficiencies and waste in the field of clinical trials, Chalmers *et al.*, suggest that research on research is key to enhancing efficacy in trial methodology [6]. An example of research on trial methodology is the Prioritising Recruitment in Randomised Trials Priority Setting Partnership Study (PRioRiTy PSP) [7], Healy *et al.* identified and prioritised unanswered questions around trial recruitment research. Following on from this work, Hennessy *et al.*, [8] list significant opportunities for qualitative methodologies to contribute to better understanding of trial recruitment issues and suggest that the true value of such methodologies is yet to be fully realised. This qualitative evidence synthesis (QES) was designed to specifically focus on question five of the PRioRiTy Study; 'What are the barriers and enablers for clinicians/healthcare professionals in helping conduct randomised trials?' within the context of recruitment during pregnancy and childbirth, but could well have learning relevant to other top 10 areas.

We conducted a scoping search to help formulate the research question and identify key search terms. This revealed limited qualitative evidence on recruitment to trials during pregnancy and childbirth from either the woman's or the recruiter's perspective, as historically, these women had been excluded from participation in clinical research studies. Among the reasons for exclusion are; the potential risk of teratogenic effects to the foetus, adverse pregnancy outcomes [9], and the 'vulnerability' of the population [10], although more recent debate challenges the notion of vulnerability [11]. Mohanna & Tunna [12] highlight the unique position posed by recruitment during pregnancy and childbirth as two people are involved, the woman and her unborn baby. How the woman views the trial and the decision to participate or not may shift according to which lens (maternal or foetal) is applied. In addition to this, the father's attitudes and views may also influence the decision-making process [12]. These combined factors have likely resulted in the underrepresentation of women during pregnancy and childbirth in clinical research [13, 14], and consequently little evidence to guide researchers on how to effectively recruit this cohort to clinical trials.

Patient experiences of recruitment to clinical trials has been the focus of previous QES [15] and one ongoing [16], both have included women during pregnancy and childbirth. However, the experiences of trial recruiters are largely absent from the literature on clinical trials during

pregnancy and childbirth. Directed by the Priority study [7] question on barriers and enablers for healthcare providers in recruiting to RCTs we attempt to address this evidence gap, to gain a better understanding of the experiences and perceptions of recruiters across a range of maternity care settings and conditions, e.g. emergency and non-emergency. To our knowledge this is the first QES exploration in this area and provides a single point of access for synthesised evidence on recruiting this cohort, which can be used to inform decisions around trial methodology.

The aim of this QES is to explore the evidence on the recruiter's experience and perceptions of recruitment during pregnancy and childbirth to trials. The specific objectives for this review are:

1. To describe recruiters' perceptions and experiences of recruiting women during pregnancy & childbirth (period extends from first trimester to six weeks postpartum) to clinical trials.

2. To describe the recruiter's perceptions and awareness of how their own role (e.g. clinical or non-clinical) may have an influence on recruitment.

3. To explore the recruiter's perceptions of how the 'type of trial' might have an influence on their experience of recruitment.

4. To explore the setting and environment in which recruitment is undertaken, and the recruiter's perception of what influence setting and environment has on their experience of recruitment.

## Methods

The protocol for this QES, was considered by the team at PROSPERO as outside the scope of PROSPERO registration, as it does not address health outcomes. However, it is publicly available online (https://osf.io/g4dt9/). The ENTREQ statement [17] was used to guide the reporting of this QES as it is recognised to improve both the conduct and reporting of qualitative syntheses and enable the end-user to better understand the processes involved in developing a qualitative synthesis (See S1 Table).

### Literature search and selection

**Inclusion criteria.** *Types of studies*. We included all qualitative studies, such as grounded theory studies, phenomenological studies, ethnographies, and case studies. Studies using qualitative methods for data collection, such as focus groups, face-to-face interviews, observations, arts-based methods or document analysis, and for data analysis such as content analysis, thematic analysis, constant comparison, or other qualitatively inspired analytical approaches were included. Editorials, commentaries, opinion papers, and studies that did not provide a transparent audit trail of the methods used were not be included. Only studies published in English were included as language nuances often present difficulty when interpreting concepts.

*Types of participants*. All studies focusing on the perceptions and experiences of recruiters when recruiting women and/or their partners during pregnancy and childbirth to clinical trials were included. No restrictions were placed on age, social status, ethnic background, or country of recruitment.

### Study search strategy and process

An expansive rather than exhaustive search [18], was undertaken. This method of literature search is appropriate to QES as it affords the opportunity to continually redefine the research

question and explores the emergence of research findings. The search strategy (See S1 Appendix) combines the five concepts of the SPIDER tool [19]:

**S**ample–Participants and Recruiters (clinical & non-clinical)

**P**henomenon of **I**nterest—Recruiting women to clinical trials during pregnancy & childbirth (clinical trials are defined as any clinical/medical research requiring consent to participate)

**D**esign—"case study" OR observation OR "focus group*" OR interview* OR

**E**valuation—experience* OR attitude* OR belief* OR perceive OR perception OR opportunit* OR opinion

**R**esearch Type—Qualitative OR "mixed-method" OR "mixed method" OR "multi-method" OR "Multi method"

**Electronic searches.**    We conducted a tailored search using a combination of index terms and free-text words, of each of the following databases:

- CINAHL (OvidSP) (1937 to 7[th] January 2019)

- Embase (OvidSP) (1947 to 7[th] January 2019)

- PubMed (OvidSP) (1948 to 7[th] January 2019)

- PsycINFO (OvidSP) (1887 to 7[th] January 2019)

Using the table of terms (S1 Appendix), synonyms for the key search terms were entered into the database and truncated where appropriate. The individual searches for each SPIDER component were combined using the "OR" Boolean operator into a single group. Initially, the groups S and PI were combined with "AND" and then, in turn, "AND-ed" with the three methodological terms (namely "D OR E OR R") as suggested by Cooke *et al.*, [19]. The final list of citations were saved into Endnote (Thomson Reuters 2012) and screened for duplicates. Records of all searches in each database were maintained [20].

**Searching other resources—grey literature.**    Relevant resources were identified through backward reference and forward citation searching of reference lists using Scopus, Web of Science and Google Scholar. In addition, other sources such as reports, dissertations, theses databases and databases of conference abstracts (ETHOS, ProQuest) were searched. Hand searches of relevant journals, authors of relevant studies and content experts were performed looking for additional published or unpublished work. Google Books was also searched to retrieve books and book chapters.

## Study selection

All articles underwent a two-stage screening process (Fig 1). In stage 1: all citations were screened based on title and abstract. Rayyan QCRI software [21] was used by two review authors (VH & LB), independently to determine if the predefined set of inclusion criteria outlined by SPIDER [19] were met. In stage 2: full text of each included article was obtained, read in full and assessed for inclusion independently by both reviewers (VH & LB). Authors met regularly to discuss any differences in opinion were resolved through discussion.

## Appraisal of the methodological limitations of included studies

The methodological limitations of the included studies were appraised using an adapted version of the Critical Appraisal Skills Programme [22]. This version, used by Ames *et al.*, [23] and more recently in Karimi-Shahanjarini *et al.*, [24], offers a reasonable framework to assess the methodological limitations of primary qualitative studies in the absence of a gold standard approach.

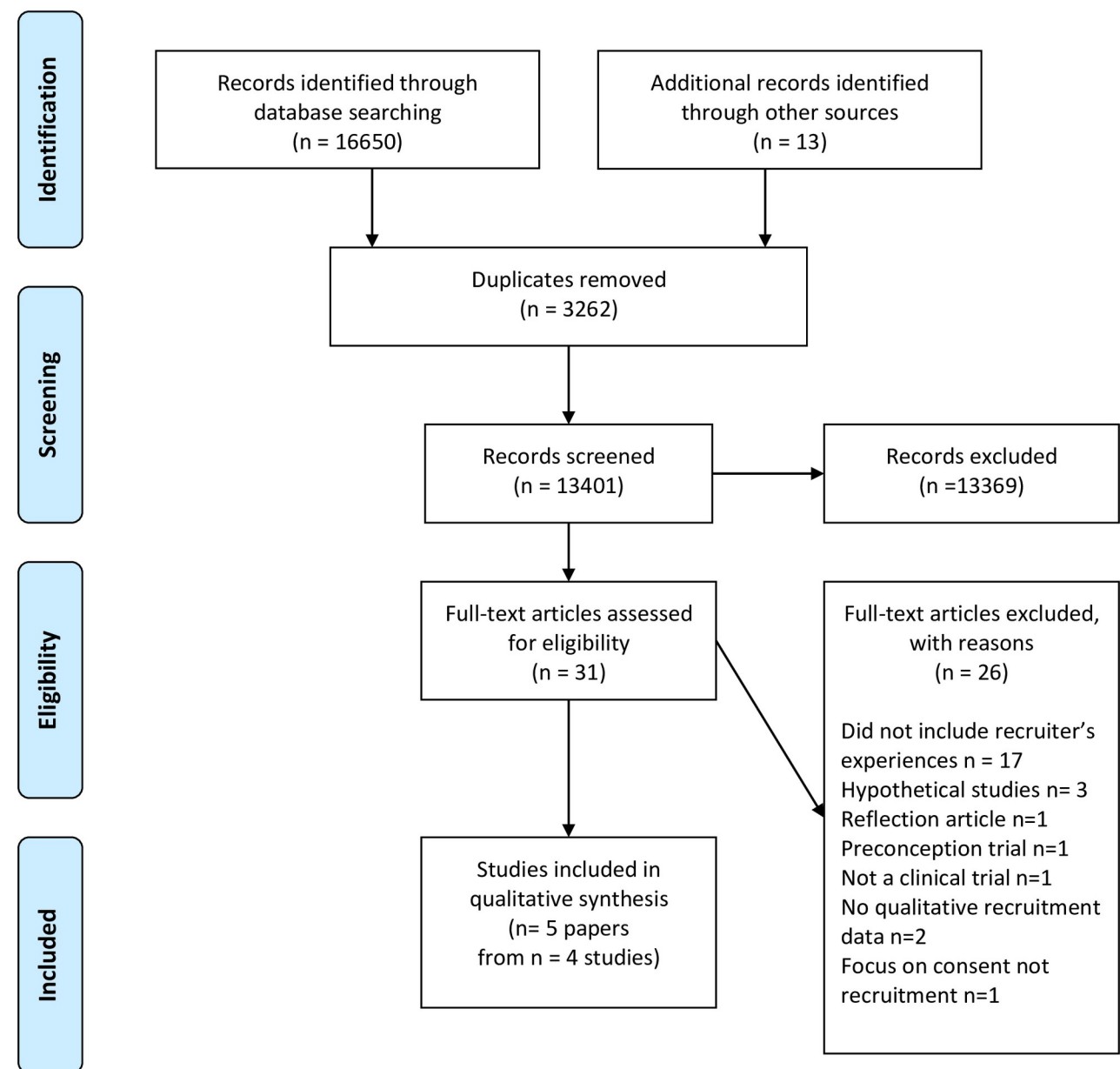

*From:* Moher D, Liberati A, Tetzlaff J, Altman DG, The PRISMA Group (2009). *P*referred *R*eporting *I*tems for *S*ystematic Reviews and *M*eta-*A*nalyses: The PRISMA Statement. PLoS Med 6(7): e1000097. doi:10.1371/journal.pmed1000097

**For more information, visit www.prisma-statement.org.**

**Fig 1. PRISMA flow diagram.**

The adapted tool includes the following eight questions:

1. Are the setting(s) and context described adequately?

2. Is the sampling strategy described, and is this appropriate?

3. Is the data collection strategy described and justified?

4. Is the data analysis described, and is this appropriate?

5. Are the claims made/findings supported by sufficient evidence?

6. Is there evidence of reflexivity?

7. Does the study demonstrate sensitivity to ethical concerns?

8. Any other concerns?/Overall assessment of methodological limitations

## Quality assessment

Two authors (VH, LB) assessed the quality of included studies independently, appraisal was compared, and agreed by consensus (See S2 Table). The summary of methodological limitations is included in Table 2. No studies were excluded on the basis of quality assessment, rather the quality assessment served as an identifier of weak study methodology and facilitated the later confidence of findings assessment [25]. The Cochrane Qualitative and Implementation Methods Group, whilst acknowledging the subjectivity of critical appraisal, comment that the guiding principle of undertaking assessment of methodological strengths and weaknesses is transparency in the reporting of all decisions and to provide rationale to support them [26].

## Public and patient involvement

This QES had no direct patient and public involvement but there was indirect involvement through other sources. The PRioRiTy Study [7], incorporated public and patient involvement (PPI) from inception to dissemination ensuring it's meaningful and relevant contribution. This QES has advanced the work of Healy and colleagues [7] by synthesising the existing qualitative studies in relation to recruiter's experiences and perceptions of recruitment during pregnancy and childbirth to trials.

## Reflexivity

Reflexivity is a key component of qualitative research, and its use enhances the audit trail by providing records of personal responses and contributions of the review author and the larger team [27]. In keeping with quality standards for rigour in qualitative research, we considered our views and opinions on recruitment to clinical trial during pregnancy and childbirth, and the possible influences on the decisions made in the design and conduct of this QES, including the search strategy, inclusion decisions, synthesis and interpretation of the findings. VH, is a registered midwife, and has experienced the barriers clinical staff encounter when recruiting for trials and came to the project with prior beliefs about the complexity and interdependency of factors which impact the recruitment of pregnant women to clinical trials. LB, a lecturer of midwifery, was also informed by a clinical background in midwifery, is experienced in leading previous primary research with women and health professionals in Ireland. KG, is a clinical trial methodologist with a specific interest in how people make decisions about participation in clinical trials. The resolution of any disagreements regarding decisions made throughout

the selection and review process has been documented, in the form of a reflexive journal, and forms part of the written audit trail.

## Data abstraction and synthesis

Guided by the RETREAT framework [28], we undertook this qualitative evidence synthesis using a thematic synthesis method by Thomas & Harden [29]. This method was chosen as it draws on the conceptualisation and techniques of thematic analysis in order to facilitate the transparent identification and development of descriptive and analytical themes. Given the small number of papers, we used MS Excel to manage the data and opted to code and synthesise data manually. The thematic synthesis included three overlapping stages: line by line coding, developing descriptive themes, and generation of analytical themes. Papers were ordered alphabetically, and data extracted in order. All qualitative findings from the primary studies relevant to the research question were extracted. Findings were defined as any qualitative data describing a theory, new concept, theme, sub-theme or finding statement, presented in forms including, but not limited to, text, tables, diagrams, supplementary files located anywhere in the paper. Through a line-by-line coding process (VH) we developed 'free codes' (without hierarchical structure), this bank of codes grew as each paper was coded. An iterative process was then applied as data was examined and checked (VH, LB) for meaning and content, this process allowed the translation of codes and concepts between studies. As similarities and differences between codes became apparent, we grouped them into a hierarchical structure to form descriptive themes (VH, LB). Finally, we attempted to generate analytical themes from each category to go beyond the findings of the original studies into a higher order abstraction of the phenomena (VH, LB, KG) (see S3 Table). We looked for agreement and differences in views and perceptions of recruiters within and across settings such as hospital and community, recruiters professional and clinical backgrounds, and different types of trials.

## Assessment of confidence in the review findings

We applied Grading of Recommendations Assessment, Development and Evaluation-Confidence in the Evidence from Reviews of Qualitative research (GRADE-CERQual) [30] to the findings of the thematic synthesis. The GRADE-CERQual approach is based on four components which include: the methodological limitations of included studies, the coherence of the review findings and the adequacy of data contributing to the review findings and the relevance of the included studies to the review question. A 'Summary of Qualitative Findings' was derived from the findings of the thematic synthesis, as the research team summarised and identified the central idea of each finding, the authors then made an overall GRADE-CERQual assessment of confidence based on these findings. Each review finding was assessed by review authors (VH, LB), concerns regarding any of the four components was noted. Findings were discussed with the third author (KG) before applying the overall GRADE-CERQual assessment. We based our judgements on an initial assumption that all findings were 'high confidence' and were a reasonable representation of the phenomenon of interest, and then downgraded them accordingly if there were concerns regarding any of the GRADE-CERQual components. The 'Summary of qualitative findings' along with their GRADE-CERQual rating are shown in Table 2, in addition, the detailed 'Evidence profile' for each finding is shown in S4 Table.

## Findings

This synthesis of qualitative evidence includes five papers, from four studies, all of which were identified through an electronic database search. Four are from the United Kingdom [31–34]

and one from The Netherlands [35]. Four of the studies were hospital based and involved trial recruitment in an emergency or time-critical situation [31–33, 35], of these one was study involved a non-medicinal intervention [31], while the other three had CTIMP interventions [32, 33, 35]. Stuart *et al.'s* study was in a primary care setting, non-emergency, with a non-CTIMP intervention [34]. Most data were collected by individual interviews and/or focus group discussion. Table 1 details the characteristics of the studies included in the review.

Sample sizes ranged from 13 to 27 (median = 20), with a total of 71 participants represented in this QES, two studies [32, 33] contained data from the same sample. Contributing to the sample were: clinical midwives (n = 28), research midwives (n = 11), obstetricians & gynaecologists (n = 19), neonatologists & paediatricians (n = 8) and neonatal nurses (n = 5).

Sixteen summary of qualitative findings statements (SQF) were gleaned from the synthesised evidence, these mapped onto four distinct themes (Fig 2): recruitment through a clinician's lens (four SQF); recruiter's judgement of acceptability (six SQF); from protocol to recruiter's lived experience (four SQF); and framing recruitment in context (two SQF) (see Table 2). Exemplars of supporting data for each theme are provided in S5 Table.

## Theme 1: Recruitment through a clinician's lens

Recruiters, across all of the studies included for synthesis, were trained clinicians by background, however some (n = 60) remained in clinical roles providing maternity care, and other (n = 11) held research roles. The range of their professional backgrounds included midwives, neonatal nurses, neonatologists, obstetricians and gynaecologists (Table 1). Overall, trial recruitment relied on the cooperation of clinicians to undertake recruitment in addition to their clinical role. It was not clear from the studies whether clinicians had a choice or not in assuming this additional role or if it was an expectation within their clinical environment. Undertaking the role of recruitment was placed within the wider context of clinical duties, and as such, the task was viewed from a clinician's perspective. This theme focuses on how the clinical role identity of the recruiter influenced their experiences of recruiting women to clinical trials.

**1.1 Clinical care is the priority.** The duality of the roles of clinician and recruiter led to competing considerations in the management of the recruiter's workload. In all five of the studies in this synthesis, there was evidence that recruiters prioritised clinical care. Two studies [32, 34], were explicit in reporting *"research was not a priority"* and that recruiters considered recruitment secondary to clinical care.

Recruiters, commonly reported experiencing 'pressure' and feeling 'burdened' by the additional task of recruitment. This pressure was often attributed to managing a high clinical workload and staff shortages, which made their involvement in recruitment difficult.

In the majority of studies [31–34], recruiters emphasised that their busy clinical workload did not allow time for recruitment. This sentiment was in complete contrast to the experiences of recruiters from the same study whose role was dedicated recruitment [31].

Whilst all of the recruiters across the studies included in this synthesis, engaged in their trial recruitment role, they primarily identified with their clinical role, and for the variety of reasons mentioned, prioritised the woman's clinical care over their recruiting responsibilities.

**1.2 Recruiter's perception of pregnant women in clinical trials.** The clinical background of the recruiter influenced their perception of pregnant women's participation in clinical trials. Recruiters made judgements, based on their clinical assumptions, about a potential participant's mental capacity (or lack of,) which determined, whether or not they engaged them in the recruitment process, and if so, how they went about it [31, 33–35]. These judgements were

**Table 1. Characteristics of included studies.**

| Author & Year | Country & setting | Type of trial/ intervention | Aim | Study populations & sample size | Methodology/ design | Key Themes | Limitations |
|---|---|---|---|---|---|---|---|
| Chhoa *et al.*, 2017 [31] | England Maternity hospital | Cord Pilot Trial (immediate or deferred cord clamping in very preterm birth | To assess clinicians' views and experiences of offering these two consent pathways. | Clinicians n = 7 *N = 5 consultant neonatologists, N = 3 neonatal or paediatric registrars, N = 5 neonatal nurses, N = 4 midwives* | Qualitative Interview—open ended questions Inductive thematic analysis | Six Themes (1) team approach to offering participation (2) consent form as a record (3) consent and participation as a continual process (4) different consent pathways for different trials (5) balance between time, information, and understanding (6) validity of consent | Results are based on a single trial, and other factors may be more important, or less important, in trials with different populations and different risk and benefit profiles. |
| Hallowell *et al.*, 2016 [32] | England Maternity hospitals (8 pilot sites) | Got-it trial is a randomised, placebo-controlled, double-blind, pragmatic UK-wide randomised controlled trial (RCT) involving women who have a retained placenta (RP) recruited from delivery wards in UK maternity hospitals. | To explore staff's and women's experiences of, and views about, the information and consent pathway used in the pilot phase of the Got-it study. | Staff n = 27 *including clinical midwives, research midwives and obstetricians (consultants, registrars, specialist trainees)* | The study was informed by the principles of grounded theory Simultaneous data collection and analysis Data were collected during telephone interviews (with one exception) | Four Themes Accounting for uncertainty: initial perceptions of trial design Familiarity breeds therapeutic optimism: perceptions of the study drug Constructing therapeutic optimism: perceptions of clinical need Maintaining therapeutic optimism and sustaining trial recruitment | Confined to clinical staff who were involved in Got-it; therefore, it does not provide insight into the views of those who had declined to recruit to this trial Self-presentation; it is possible that the staff we interviewed tailored their accounts of trial involvement to present themselves as responsible and morally upstanding individuals. Interviews were carried out during the first few months of the Got-it trial; one might, therefore, expect high levels of TO to be reported at this point. Data from staff involved in one trial. staff feel and act very differently about different trials, difficult to generalise from these results. |

*(Continued)*

**Table 1.** (Continued)

| Author & Year | Country & setting | Type of trial/ intervention | Aim | Study populations & sample size | Methodology/ design | Key Themes | Limitations |
|---|---|---|---|---|---|---|---|
| Lawton *et al.*, 2016 [33] | England Maternity hospital | Got-It Trial (see above dscription) | To explore the experiences and views of women and trial staff about the information and consent pathway used within the pilot with close attention paid to the potentially challenging context within which recruitment took place. | Women n = 22 <br><br> Staff n = 27 *(n = 10 Doctors, n = 3 consultants, n = 6 Clinical MW, n = 2 Labour ward Lead, n = 11 Research MW)* | Qualitative Study Design <br><br> In-depth interviews <br><br> Iterative approach that entailed simultaneous data collection & analysis <br><br> Interviews of women and staff in parallel <br><br> Thematical analysis using method of constant comparison | Seven Themes <br><br> 1) Women's views about the trial <br><br> 2) Women's views about information delivery and giving informed consent <br><br> 3) Women's experiences of, and views about, the information and consent pathway <br><br> 4) Women's views about improving the information and consent pathway: extending information-giving into the antenatal period <br><br> 5) Women's views about improving the information and consent pathway: extending information-giving into the postnatal <br><br> 6) Staff views and experiences of recruitment and gaining informed consent <br><br> 7) Staff views about the timing of information delivery and extending the consent pathway | Could not evaluate whether, and how, pregnant women may be sensitised to the issues around RPs. <br><br> Powerful endorsement for information to be given at the time of recruitment in simplified verbal and written forms, the relatively straightforward (and hence easy to explain) nature of the Got-it trial intervention needs to be taken into account. |

*(Continued)*

**Table 1.** (*Continued*)

| Author & Year | Country & setting | Type of trial/ intervention | Aim | Study populations & sample size | Methodology/ design | Key Themes | Limitations |
|---|---|---|---|---|---|---|---|
| Stuart *et al.*, 2015 [34] | England Community Setting *(midwives from 6 of the 7 sites participating in the RCT (UK NHS Group Family Nurse Partnership Programme))* | First Steps Randomised Controlled Trial of Group Family Nurse Partnership (gFNP). The multisite RCT aimed to examine if provision of gFNP, compared to routine antenatal and postnatal services, could reduce risk factors for child maltreatment (Barnes et al., 2013) | To investigate the perceptions of community midwives about their role in identifying potential participants in early pregnancy for the first steps RCT trial of Group Family Nurse Partnership (gFNP). | Community Midwives n = 13 | Descriptive qualitative investigation Face-to-face, semi-structured, using interview Thematic content analysis | <ins>Five Themes</ins><br><br>1) Issues with the midwifery role in the trial process<br><br>*(a) Insufficient information about their role*<br><br>*(b) Recruiting versus identifying trial participants*<br><br>*(c) Insufficient time for research activities*<br><br>2) Issues with the criteria for trial participants<br><br>*(a) Difference between identifying for FNP and gFNP*<br><br>*(b) Criteria too detailed*<br><br>*(c) Not a fit with their population*<br><br>3) Reasons for potential participant refusal<br><br>*(a) Pregnant women dislike groups*<br><br>*(b) Will not be able to travel to groups*<br><br>*(c) Targeting vulnerable group who do not want more support*<br><br>4) Reservations about midwifery care as part of gFNP<br><br>*(a) Concern about the midwifery care and system breakdown*<br><br>*(b) Concerns about self-care*<br><br>5) Views about the gFNP programme in the future<br><br>*(a) Potential for positive impact*<br><br>*(b) Possible identification in future mainstreaming* | Small number of respondents may not represent all experiences, which would have been possible with a larger sample if the study had been funded to explore midwives' experiences in more depth. |

(*Continued*)

**Table 1.** (Continued)

| Author & Year | Country & setting | Type of trial/ intervention | Aim | Study populations & sample size | Methodology/ design | Key Themes | Limitations |
|---|---|---|---|---|---|---|---|
| van der Zande *et al.*, 2019 [35] | Netherlands Two academic hospitals | APOSTEL VI study assesses whether a cervical pessary prolongs pregnancy in women who have been admitted for threatened preterm birth but remained undelivered after 48 hours | To explore what stakeholders think about inclusion of pregnant women in the APOSTEL VI study: a low-risk obstetrical randomised controlled trial (RCT). | Healthcare professionals n = 14 *(Gyn/Ob n = 9, Mw n = 5)*<br><br>REC members n = 5<br><br>Regulators n = 5<br><br>Pregnant Women n = 14 | Qualitative study design<br><br>Semi-structured interviews<br><br>One focus group discussion with Regulators | <u>Four Themes (3+1)</u><br>1) Motivations for participation<br>2) Gatekeeping<br>3) Counselling<br>4) Interest in (routine) inclusion | 1) Sample highly educated stakeholders regarding only the Dutch situation.<br><br>2) Saturation number of 20 interviews was reached on group level, but not always on sub-group level.<br><br>3) Only included pregnant participants who were recruited for the APOSTEL VI study, a group that consists of women that become sick during their pregnancy and who are recruited for a low-risk obstetric study.<br><br>4. No data from any representatives from a pharmaceutical company. |

based on their perceptions that pregnant women lacked the capacity to understand and make decisions around trial information.

The recruiter's assumptions around women's educational attainment was a factor in three studies [31, 33, 34]. Recruiters were concerned that the women, they perceived to have lower educational levels, would not be able to understand what trial participation involved [31]. Furthermore, in two of the studies recruiters expressed the view that potential trial participants did not have the ability to read or were illiterate [33, 34]. In many of the studies, recruiters regarded pregnant and child-birthing women as vulnerable and as such, expressed the need to protect them from research [31, 33–35].

The recruiter's experiential knowledge as a clinician influenced their perception of how receptive pregnant women would be towards the trial [34, 35]. For example, midwife recruiters in Stuart's study [34], assumed that women would be disinterested in trial involvement, and because of this, despite meeting the inclusion criteria of the trial, they did not invite them to participate.

The recruiter's perception of pregnant women was also apparent in the language they used when referring to potential participants. In two studies [34, 35], recruiters used language that implied ownership of the pregnant women in their care. For example, recruiters refer to women as '*theirs*' [35], while two midwife recruiters in another study [34] are quoted referring to women as *"my ladies..."* and *"our girls"*. This suggests that recruiters assumed a deep knowledge of potential participants way of thinking.

From the studies included in this qualitative evidence synthesis, it is evident that recruiters could not ignore their maternity care professional status. It appears that their professional knowledge was an influencing factor in how they viewed potential participants and the task of recruitment, and consequently shaped the manner in which they went about it.

**Table 2. GRADE-CERQual summary of qualitative findings.**

| Summary of Review Finding | Studies Contributing to the Review Finding | CERQual Assessment of Confidence in the Evidence | Explanation of CERQual Assessment |
|---|---|---|---|
| Theme 1: Recruitment through a clinician's lens | | | |
| 1. Recruiters in the maternity care setting were predominately clinicians. Primarily identifying as clinicians, recruiters prioritised clinical care over recruitment responsibilities. | [31–35] | Moderate confidence | Minor methodological concerns with four studies as reflexivity is not addressed. Moderate concerns about the fit with data in the primary studies. Minor concerns about adequacy as studies 2 & 3 based on same participants. |
| 2. The duality of the clinician/recruiter role meant that recruiters were already busy juggling a high clinical workload, which allowed little time for the task of recruitment. In contrast, recruiters employed purely in a research capacity, experienced recruitment as unhurried and complementary to the women's care experience | [31, 32, 34] | High confidence | Only minor methodological concerns with all three studies as reflexivity is not addressed. |
| 3. Recruiters made judgements, and assumptions, about pregnant women's mental capacity and their ability to comprehend what trial participation involved. Recruiters regarded pregnant women as vulnerable and as such they believed women needed to be protected from research and therefore were less inclined to recruit them to trial. | [31, 33–35] | High confidence | Only minor methodological concerns with all four studies as reflexivity is not addressed. |
| 4. Recruiters made assumptions of knowledge regarding how receptive pregnant women would be towards the trial, and also demonstrated an 'ownership' of women through the language they used. | [34, 35] | Moderate confidence | Minor methodological concerns with both studies as reflexivity is not addressed. Moderate concerns regarding coherence and adequacy, however, both studies offered rich data. |
| Theme 2: The recruiter's judgement of acceptability | | | |
| 5. Recruiters made a judgement on the clinical relevance and aim of the trial. Concordance between the recruiter's values and beliefs, and the research goals, promoted their willingness to engage in recruitment. Whilst discordance, in turn, dissuaded recruitment. | [32–35] | Moderate confidence | Minor methodological concerns with four studies as reflexivity is not addressed. Moderate concerns regarding coherence and minor concerns about adequacy as studies 2 & 3 based on same participants. |
| 6. It was important to recruiters that recruitment processes and protocols were pragmatic and an efficient use of the resources available. | [33, 34] | Moderate confidence | Minor methodological concerns with both studies as reflexivity is not addressed. Moderate concerns about adequacy as two studies with moderately rich data. |
| 7. Recruiters formed an opinion around the acceptability of the trial intervention based on its utility, potential benefits for stakeholders (women, clinicians, and the organisation of care), and potential to ultimately improve the current standard of care. | [32–35] | High confidence | Only minor methodological concerns with all four studies as reflexivity is not addressed. |
| 8. Recruiters expressed optimism and hope for a successful trial outcome. Their engagement triggered a cyclical process where the recruiters 'buy in' to the trial generated recruitment, leading to a sense of achievement, which in turn provided positive reinforcement for their efforts and generated further recruitment. However, the cycle could also be reversed when declining recruitment rates lead to disengagement. | [32–34] | Moderate confidence | Minor methodological concerns with all three studies as reflexivity is not addressed. Moderate concerns about coherence and adequacy as studies 2 & 3 based on same participants. |
| 9. Recruiters had strong ties with established clinical practice and were uncomfortable recruiting for a trial that moved away from their routine. | [34, 35] | Low confidence | Minor methodological concerns with both studies as reflexivity is not addressed. Very minor concerns about coherence. Serious concerns about adequacy as both studies offered thin data. |
| 10. The recruiter's perception of risk (associated with the intervention) was fundamental in their judgement of acceptability, and therefore a key determinant in their willingness to engage in recruitment. Recruiters were more comfortable recruiting to a trial they considered to be low risk. | [31, 32, 34, 35] | High confidence | Minor methodological concerns with four studies as reflexivity not addressed. Only minor concerns with coherence and adequacy. |
| Theme 3: From protocol to recruiter's lived experience | | | |

*(Continued)*

**Table 2.** (Continued)

| Summary of Review Finding | Studies Contributing to the Review Finding | CERQual Assessment of Confidence in the Evidence | Explanation of CERQual Assessment |
|---|---|---|---|
| 11. Recruiters were also gatekeepers to potential participants and assumed the role of a protective advocate of women. Recruiters were paternalistic in the role and were seen to withhold trial information and steer women towards decision making which was aligned with their own views. Gatekeeping could also include protecting the interests of the trial | [31–35] | High confidence | Minor methodological concerns with five studies as reflexivity is not addressed. Only minor concerns with coherence and adequacy. |
| 12. An additional layer of gatekeeping existed between midwife recruiters and recruiters from other professional backgrounds. In what was apparently a culturally appointed hierarchy, recruiters sought permission from the midwife (recruiter) to approach a potential participant. | [31, 34] | Low confidence | Minor methodological concerns with both studies as reflexivity is not addressed. Serious concerns about adequacy as only two studies offered thin data. |
| 13. Recruitment frequently involved a team approach, with often more than one recruiter over several encounters. Recruiters often engaged in an 'exit' encounter, post-trial, which appeared to bring closure to the trial recruitment process for both women and recruiters. | [31–33] | Moderate confidence | Minor methodological concerns with three studies as reflexivity is not addressed. Minor concerns about coherence and moderate concerns about adequacy as two studies offered moderately rich data. |
| 14. There was no consensus reached amongst recruiters regarding either, the best method for communicating trial information to potential participants or at what time point the information should be given. | [31, 33] | Moderate confidence | Minor methodological concerns with both studies as reflexivity is not addressed. Minor concerns about coherence and moderate concerns about adequacy as both studies offered thin data. |
| Theme 4: Framing recruitment in context | | | |
| 15. Recruiting pregnant women during an emergency or in a time-critical situation challenged recruiters to communicate effectively in a highly pressured time constrained environment. Recruiters were not comfortable with the task of obtaining informed consent in this environment. | [31, 33] | Moderate | Minor methodological concerns with both studies as reflexivity is not addressed. Minor concerns with both coherence and adequacy as moderately rich data offered. |
| 16. Training in methodological aspects of the trial and recruitment protocols provides recruiters with the knowledge and confidence to perform their recruitment task well. Recruiters recognised the need for regular structured multidisciplinary recruitment training, both initially, and throughout life of trial | [31, 32, 34, 35] | High confidence | Only minor methodological concerns with four studies as reflexivity is not addressed and minor concerns with adequacy as one study offered moderately rich data. |

Table format from Lewin *et al.*, 2018 [36]

## Theme 2: The recruiter's judgement of acceptability

This theme focuses on the recruiter's perception of the acceptability of both the trial and the intervention. Acceptability was founded on the recruiter's opinion of the trial aim, methodology, and proposed processes (e.g. recruitment pathway), in addition to their opinion on the acceptability, or not, of the trial intervention itself.

**2.1 Acceptability of the trial.** It was widely recognised that the clinical relevance and nature of the research was important to recruiters and an influential factor in how they viewed the trial [31, 33–35]. Recruiters were reluctant to seek trial participants if they deemed the research study was of little relevance to the women to whom they offered care [35]. Recruiters were curious to understand the aim and the rationale behind the trial, which assisted them in forming an opinion on the acceptability of the trial [32]. In determining the acceptability of the trial, recruiters needed their views to be in synchronicity with the aim of the trial and that

Themes and sub-themes
of thematic synthesis

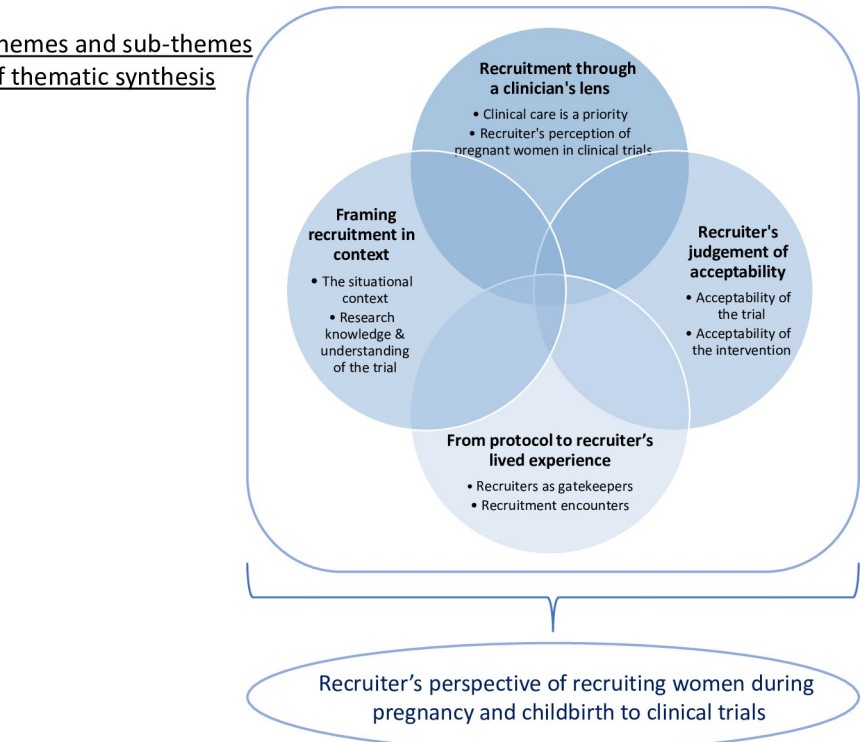

**Fig 2. QES framework.**

the trial made astute use of scarce resources [34]. Furthermore, the recruiter's judgement of trial acceptability also centred on the design of the trial processes such as the information and recruitment pathways. At times the recruitment protocols themselves were considered a hinderance to the recruitment, for example, eligibility criteria that was considered too stringent presented an obstacle for some recruiters [34]. While, recruitment protocols that made efficient use of time and resources were also fundamental to the recruiter's judgement of acceptability [33].

The acceptability of the trial may also rest on the recruiter's perception of what impact the trial outcome could have on their way of practice. Interestingly, authors of one study noted that recruiters were less likely to seek participants, if they considered the trial had the potential to undermine their clinical practice [34]. While judging the nature and methods of the trial was important to recruiters, so too was their assessment of the intervention itself, which proved equally important in their overall opinion on acceptability.

**2.2 Acceptability of the trial intervention.** In several studies, recruiters took a pragmatic view of the usefulness of the trial intervention [31, 33–35]. The recruiters emphasised that it was important to them that the trial intervention had the potential to contribute to an improvement in care. The recruiter's perception of the utility of the trial intervention appeared to be measured by several criteria, e.g. whether the potential outcome addressed an unmet clinical need [35], and whether receiving the trial intervention enhanced the woman's care experience and, was of therapeutic benefit [33].

Recruiters also appeared to consider the trial intervention's utility beyond improving care for women, e.g. identifying potential benefits of its use for clinicians, recruiters, and the organisation of care [32].

The recruiter's judgement of acceptability of the trial intervention was confirmed by their 'buy-in' and enthusiasm for recruitment. Across three studies, recruiters shared their optimism for a successful outcome [32, 34, 35], which appeared to incentivise their recruitment efforts.

The phenomenon of 'therapeutic optimism', (namely, hoping that one will benefit from trial participation), was described in one of the studies as being tied to recruiter morale [32]. Suggesting that, in trials where recruiters witnessed an outcome that did not match their expectations, recruiters were less inclined to engage with recruitment. If recruiters judged the intervention not to be acceptable, they were less likely to recruit. In one study [35], recruiters showed reluctance to recruit to the trial because the intervention required a change in practice which contravened routine care. Recruiters were also less inclined to recruit to the trial if they considered the intervention suboptimal to existing care offered and expressed concern that trial participants could *"be lost or missed"* due to receiving the intervention [34].

Judgment on the acceptability of an intervention was also dependent on the recruiter's perception of the risks involved in receiving it [31, 32, 34, 35]. Some recruiters were explicit in expressing their concern about the issue of safety for the women receiving the intervention [34].

In contrast, depended on the nature of the trial, recruiters believed there to be little to no risk in receiving the trial intervention they were involved with [31, 32].

In studies where the trial recruiter also delivered the trial intervention [31, 32], the concept of risk appeared especially important. The authors of one study remarked that because recruiters were familiar with the intervention, they more comfortable recruiting to and delivering a trial intervention as they perceived it to be low risk [32]. However, the same study also noted that as recruiters witnessed the apparent trial outcome, their perception of risk (associated with the intervention) could shift during the life of the trial [32].

The recruiter's judgement on acceptability appears to centre around; their beliefs about the clinical relevance and utility of the trial, their ability to navigate the information and recruitment pathways, the utility of the intervention and their perception of risk associated with participating. When recruiters found these aspects acceptable, they were more likely to engage with recruitment to the trial.

## Theme 3: From protocol to recruiter's lived experience

The third theme focuses on the recruiter's lived experiences and interactions with trial recruitment on a practical level. The synthesis of evidence reveals a discordance in the operationalisation of trial recruitment between the theoretical plans and the actual lived experience of delivering the recruitment phase of the trial. This theme considers how the recruiters self-perception as gatekeeper, and their recruitment encounters impacted on their experience of recruitment.

**3.1 Recruiters as gatekeepers.**   In all five studies, the recruiters also performed the role of clinical gatekeeper (i.e. controlled access to the trial) of potential trial participants [31–35]. The recruiters regarded their position of gatekeeper as advantageous for recruitment because they often had an established rapport with potential participants [31, 34, 35].

The significant contribution of the role of gatekeeper was explicitly acknowledged as essential to the recruitment process [32, 34]. Interestingly, one study observed an additional layer of gatekeeping existed between midwife recruiters and recruiters from other professional backgrounds. In what is apparently a culturally appointed hierarchy, recruiters sought permission from the midwife (recruiter) to approach a potential participant [31]. In their gatekeeper role,

recruiters often assumed a protective stance towards pregnant women, by deliberately withholding trial information [33].

There is evidence that the recruiters made the conscious decision not to follow stipulated recruitment protocol and steered women towards making a decision that aligned with their own professional opinion [35]. Recruiters felt vindicated in their decision not to include all eligible women in the trial, as they were acting '*in their patient's interests*' or did so in an attempt "*to protect the woman and her foetus and not overburden her*" [33, 35]. One study author offered a potential explanation for the lack of adherence to protocol, suggesting clinicians may experience a loss of professional autonomy in the role of recruiter which triggered gatekeeper behaviour [34].

An additional aspect of gatekeeping was highlighted in van der Zande's study, whereby recruiters also acted as gatekeeper for the trial. Some recruiters appeared protective toward the trial, and deliberately avoided some potential participants they considered not suitable trial candidates for fear they would make a "mess" of the study [35].

**3.2 Recruitment encounters.**   The recruitment encounter did not always follow protocol design. The interactions between recruiter and potential trial participant could take on many guises, from formal counselling about the trial to a passing mention at a clinic appointment. The synthesised evidence showed the various methods of trial communication recruiters used during these recruitment encounters. Recruiters often employed a teamwork approach to trial recruitment, frequently this included two recruiters from different professional backgrounds [31, 32]. Furthermore, it was not unusual for recruiters to have more than one recruitment encounter (with the exception of emergency or time-critical trials, which are discussed in the next theme). In fact, successful recruitment of women during pregnancy to clinical trials, often necessitated several counselling encounters with different research team members [32].

The methods and merits of communication during these recruitment encounters was also important to recruiters. Some recruiters believed that counselling conversations were more conducive to decision making for women, than receiving written trial information, especially in an intrapartum context [31]. While other recruiters believed giving written trial information at the time of enrolment best served recruitment and were rigid in this view, despite learning this approach was contrary to women wishes [33]. Conversely, other recruiters were of the opinion that embedding trial information earlier in antenatal care would provide a helpful introduction for potential participants and may facilitate recruitment [31]. An unanticipated finding of the synthesis was the 'exit' encounter. By way of conclusion to the woman's participation in the trial, recruiters made follow up visits. This is not something recruiters were expected or obliged to do, however, the post-trial visit appeared to draw to a close to the trial recruitment process for both women and recruiters [31, 33].

The recruiter's lived experience of trial recruitment does not always follow protocol. The synthesis of data in this theme illuminates the impact that, the recruiter as gatekeeper, and the recruitment encounters has on trial recruitment.

## Theme 4: Framing recruitment in context

This theme encompasses both the physical and virtual environments in which recruitment to clinical trials during pregnancy and childbirth occurred. The findings from this synthesis identified the role of the recruiter is framed by the context of the situation in which recruitment occurs, and by their research knowledge and understanding of the trial.

**4.1 The situational context.**   The findings showed the importance of considering the situational context in which recruitment occurs and how this is reflected in the recruiter's experience. Care in the maternity setting may be unplanned at times, and three papers included for

synthesis are derived from trials conducted during an unplanned emergency or time-critical situation [31–33]. In these papers, recruiters identified there were a number of key differences between 'routine' recruitment and recruiting pregnant women to trial during an emergency or in a time-critical situation [31, 33].

Firstly, recruitment to perinatal trials challenged recruiters to adapt how they communicated information about the trial to women and their families. Providing trial information in an inherently pressurised environment often necessitated that recruiters to be pragmatic with the amount of information they gave [31, 33]. Due to the time-critical nature of the trial, there may be a very limited time period in which discussion and decision-making can take place. Recruiters had to strike a balance between the time available and the amount of information given to women [33]. As a consequence of recruitment during an emergency or time-critical trial, recruitment and consent became a fluid encounter. The additional task of obtaining informed consent proved uneasy for some recruiters, as they called to question whether informed consent was even possible in such a time pressured context [31]. It was important to recruiters that they provide proof of a woman's willingness to participate in the trial, emphasising that evidence of informed consent offered a means of legal protection should the validity of consent ever be challenged [31].

**4.2 Research knowledge and understanding of the trial.** The synthesised evidence highlighted that research knowledge and understanding of the trial was a factor in the recruiter's ability to successfully recruit to the trial. Possessing a general methodological awareness of trial recruitment was helpful in facilitating recruiters to carry out their role [31, 34]. Indeed, this knowledge and understanding of the methodological aspects of recruitment, was a driver for participant recruitment [31]. Furthermore, knowledge of the particular trial in question, and having a sound understanding of its recruitment protocol gave recruiters certainty in their role and allowed them to be impartial in how they presented the trial to potential participants [35]. However, recruiters were not always equipped with sufficient trial information to perform their role. Ineffective methods for transferring knowledge were seen in Stuart's study, where the trial relied on information cascading from clinical managers to recruiting staff, which resulted in misunderstanding and confusion about the trial, and the recruitment process, and consequently poor recruitment [34].

Recruiters recognise the need for structured multidisciplinary recruitment training, both initially, and throughout life of trial [31, 32, 34]. The regular rotation of staff in the maternity care space was acknowledged by recruiters as a hindrance to recruitment, and they were especially keen for on-going trial and recruitment training as a way to address the deficit [31].

In addition, training concerning the conceptional underpinnings of the trial could benefit recruiters by enabling them to recognise how clinical equipoise directs their approach to recruitment and, assist them in presenting trial participation objectively [32].

The synthesis findings showed that the recruiter's experience and ability to recruit to clinical trials is framed by the context in which recruitment occurs. The situational context and the context of knowledge and understanding are influential factors in how recruiters perform their task.

## Confidence in the findings

Our confidence in the findings of this review are based on our GRADE-CERQual [36] assessment which ranged from high to low. Findings were downgraded for methodological limitations due to the lack of reported researcher reflexivity across all studies. We typically downgraded a finding for concerns about coherence when there were some concerns about the fit between the data from primary studies. Downgrading due to data adequacy occurred

when we had concerns about the richness or quantity of the data supporting a review finding. However, no findings were downgraded because of concerns about relevance.

## Discussion

This is the first report of a qualitative evidence synthesis to focus solely on the recruiter's perspectives of clinical trial recruitment of women during pregnancy and childbirth, and as such provides in depth analysis in this context. Our discussion considers the most salient findings threaded throughout the synthesis of evidence, linked by an overarching theme which combines beliefs and power. In other words, our review suggests that when it comes to trial recruitment, recruiters believe they know 'best'. Recruiters often cite their assumed knowledge and beliefs regarding women's preferences, capabilities and personal circumstances, alongside their judgements and assumptions in relation to the trial and the trial intervention. Our synthesis also highlights other considerations for recruiters which are woven into the overarching theme of beliefs and power which relate to the operationalisation or the 'doing' of trial recruitment.

Our synthesis shows recruiters frequently adopt a position of thinking they know 'best' in making judgements and assumptions, which were often overly negative which resulted in women being excluded from trial participation. Recruiter's judgements were based on experiential knowledge, and were often harsh in nature regarding women's mental capacity and educational attainment, which denied eligible women trial access. The implications of restricting trial access has potentially far reaching consequences for the health of pregnant women and their unborn babies, and limits the ability to make evidence-based clinical decisions [13]. Several recruiters referred to pregnant women as vulnerable and deserving of protection from the demands of trial involvement. This finding is commonly reported in the literature [37, 38], and may provide an important insight in understanding the cause of the underrepresentation of pregnant women in clinical research [9]. More recently, the concept of vulnerability in pregnancy is coming under challenge in the literature [11], suggesting that pregnancy should not be considered a deterrent for inclusion in clinical research, however, this concept may not have filtered through to clinical practice. Through their beliefs and professional judgement, recruiters assessed acceptability and risk, leading them to either embrace or reject the trial, and in some cases develop suspicions around it. For instance, the trial intervention could be perceived as a threat to practice for some recruiters that had strong ties with established clinical practices and were uncomfortable recruiting for a trial that moved away from their routine.

The overarching theme of beliefs and power is perhaps best illustrated in the language of paternalism that were evident in the narrative used throughout the review. The expressions of professional ownership, such as 'my girls' are particularly striking, although frequently used in the language of maternity care [39, 40], paternalistic references such as these are demeaning to women and inappropriate in any healthcare context. Further examples of paternalism were evidenced in the gatekeeping behaviours of recruiters. Many recruiters considered it their duty to act as a protective advocate for women–acting as gatekeeper between trial and woman. Although a low confidence finding, our synthesis also revealed a gatekeeping hierarchy which placed the midwife at the top, ahead of recruiters from other professional backgrounds. Findings from other research on trial recruitment has proposed that gatekeeping behaviours in recruitment may be triggered by the recruiter's own implicit beliefs about the best interests of the patient, or in some cases, the best interests of the research [41]. While, recruiter's framed paternalistic behaviour in this context as serving a moral good [42], it must be acknowledged as problematic as it derogates the woman's capacity for self-determination and is ethically indefensible [43].

Drawing on Michel Foucault's [44] notion that power is inseparable from knowledge, our synthesis identifies the authoritative power recruiters derived from their professional knowledge. For example, the professional clinical status of the recruiter utilises an existing power dynamic in their relationship with women, which enables recruiters to assert persuasive power and influence. However, the duality of the role presented a complexity, previously identified by Lawton and colleagues [45], where the blurring of boundaries between acting on clinical opinion and following recruitment protocols, meant clinician recruiters grappled with relinquishing power. Alternative explanations in the literature point to the lack of power as being responsible for waiving from ascribed protocols. For example, Benoit *et al.*, [46] suggest in their study of the social organisations of maternity care, that midwives' resistance to protocols was a response to feeling subjugated by the system in which they work and by disregarding protocol midwives exercised their power. The 'hired hands' phenomenon, identified by Roth [47], where recruiters feel they have no stake or indeed interest in the research they are working on, was found to be responsible for nonadherence to research protocol in Dyson & Dyson's [48] investigation of midwives engagement in research activity. Similarly, recruiters from the medical profession also experienced a loss of power. A seminal review by Ross and colleagues [49] found the potential loss of clinical autonomy and decision-making power deterred doctors from engaging in clinical trials.

The overarching theme of beliefs is threaded through the 'doing' of recruitment, which was evidenced by the way recruiters acted with confidence, aligning their approach to recruitment with their own beliefs which often contradicted trial protocol. Our synthesis shows that in the maternity care setting, clinical care is frequently prioritised as the demands of a busy clinical workload often superseded recruitment work. It is worth noting that in stark contrast, recruiters employed solely in a research capacity, experienced recruitment as unhurried and complementary to the women's care experience. Our synthesis highlights that recruiters are acutely aware of their knowledge needs, typically identifying their current training and knowledge levels as inadequate. It is clear that training in methodological aspects of the trial and in recruitment protocols gave recruiters the confidence and surety to perform their task. Whilst pragmatic and efficient recruitment processes are important to recruiters, no definitive method for recruitment was identified. The considerable variation in the approach to the recruitment encounter, ranging from team involvement to timing and exchanging trial information, is justifiable given the wide range of trial designs, point on pathway and setting where trial recruitment during pregnancy can occur.

Our synthesis revealed different types of trials presented various challenges. The recruiter's discomfort with recruiting pregnant women during an emergency or in a time-critical situation centred around questioning the ethics and validity of obtaining informed consent in such an environment. These issues were also raised in a recent QES presenting the conflicts and ethical dilemmas faced by research nurses when recruiting under these circumstance, and the adverse impact this had on recruitment rates [50]. One potential solution, used in emergency paediatric trials, is deferred consent, which has proven to be an effective strategy for recruitment in high-stress environments [51].

The findings of our qualitative evidence synthesis, on recruiter's perspectives of the recruitment of women during pregnancy and childbirth, are comparable with many of the findings from other reviews of recruiter's experiences in different clinical areas [52–54]. However, unlike these aforementioned reviews, randomisation and treatment preferences of potential participants were not discussed within the papers included in our synthesis. As the overarching theme illustrated, the recruiter's knowledge and power are key, both in terms of facilitating recruitment but also, in creating barriers to recruitment through paternalistic and gatekeeping behaviours.

## Implications for practice

Our synthesis has illustrated that there are a collection of factors contributing to the recruiter's perspectives and engagement with recruitment that may be overlooked during the trial development stage. The trial design community should consider the development of trial protocols in collaboration with recruiters, this could address the gap between the recruitment protocol and lived experience. Examples of involving stakeholders at all phases of trial planning and conduct have proven effective in increasing both recruitment and retention [55]. We have illustrated that recruitment in maternity care occurs within a pre-existing power dynamic which potentially gives rise to behaviours such as gatekeeping and paternalism. In tackling the issue of gatekeeping behaviours, it has been suggested that trial designers recognise and adjust for this mitigating factor by using strategies such as a gatekeeping screening tool [42]. We believe it would also be reasonable to suggest that to improve recruitment, attention should be given to expanding the clinical trials knowledge base of recruiters. Providing comprehensive training for recruiters of all professional backgrounds, both on the operationalisation and conceptual underpinnings of the research would be particularly useful.

## Implications for future research

Interventions such as QuinteT Recruitment Intervention [56] (a two-phase intervention applied to RCT's encountering recruitment difficulties, which gathers evidence at the clinical site(s) about the recruitment processes and then produces a plan to address these difficulties), have been developed to optimise recruitment and address the knowledge training gap. However, QRI has not been applied to trials in the maternity care setting and therefore further research is needed to explore the application of such interventions in the maternity care space. Given the paucity of evidence in this area, there is a need for good quality, rigorous primary studies focusing on the recruiter's perspective across various maternity contexts including multicentre trials in emergency and non-emergency situations, both CTIMP and non-CTIMP. On reflection of our findings, future research addressing other questions identified by the PRioRiTy Study [7] including; "What are the best approaches to ensure inclusion and participation of under-represented or vulnerable groups in randomised trials?"; and "What are the best approaches to ensuring manageable workloads for clinical and research staff responsible for recruiting members of the public to randomised trials?", appear particularly pertinent in gaining a richer understanding of recruitment in maternity care trials. Furthermore, there is a clear need for future primary qualitative research to include evidence of reflexivity, to provide context to, and a deeper level of understanding of the interpretations and analysis made.

## Strengths and limitations

The themes outlined in this review are a combination of descriptive and, at times, reaching into analytical level. Our ability to advance to a truly analytical level was somewhat curtailed by the number of studies (four) and therefore data available to the authors. A potential strategy to address this could be to seek extensive stakeholder input and the inclusion of a theory development stage to 'bridge the gaps' with new hypotheses. Or alternatively, investigate if any of the primary studies were linked to interventions (i.e. trial sibling studies), granting the opportunity to make use of the data in its original context to explain how a specific trial recruited and the potential impact of staff behaviours on recruitment rates. A strength of our review is the comprehensive and systematic literature search, which identified a small number of qualitative studies of trial recruitment during pregnancy and childbirth (in both non-emergency and emergency settings) from the recruiter's perspective. Following decisions made at the protocol stage, we limited the review to studies published in the English language. As the five

papers in this review represented only four studies, it is possible there may have been some over representation of particular concepts and ideas. The studies included were from UK and Netherlands, therefore it should be noted these findings are reflective of a European perspective and are from two countries with universal healthcare. We are also cognisant that the available studies did not cover the full duration of pregnancy 0r cover the full variety of care settings, and therefore not all concepts may be adequately represented in this analysis. Nonetheless, a strength of the study was the team approach undertaken for the qualitative synthesis, and the rigorous and transparent adherence to reporting standards. We acknowledge that the gender and make-up of the research team (two midwives and a trial methodologist) working in Ireland and Scotland is likely to have influenced thematic synthesis. Undoubtably, our interpretation of the primary studies and approach to synthesis is a reflection of our professional and personal backgrounds. However, we consider our reflexivity throughout the data analysis process, has strengthened the validity and rigor of the QES.

## Conclusion

Our QES is the first study to have synthesised primary qualitative evidence, from the recruiter's perspective when recruiting women during pregnancy and childbirth to clinical trials. We have gained important new insights, that go beyond the sum of parts, in identifying the overarching theme combining belief and power, which underpins the experiences and perceptions of recruiters. The synthesised evidence shows there is a gap between the trial design study protocol and the recruiter's lived experience of actualising of it. In focusing on strategies such as collaborative trial design, mitigating gatekeeping behaviours, and ensuring regular structured training in all aspects of clinical trial recruitment, recruiters may be supported and become more successful in their endeavour.

## Supporting information

**S1 Table. ENTREQ checklist.**
(DOCX)

**S2 Table. Critical appraisal skills programme.**
(DOCX)

**S3 Table. Themes matrix.**
(DOCX)

**S4 Table. CERQual evidence profiles.**
(DOCX)

**S5 Table. Exemplars of supporting data.**
(DOCX)

**S1 Appendix. SPIDER tool and search strategy.**
(DOCX)

## Author Contributions

**Conceptualization:** Vivienne Hanrahan, Katie Gillies, Linda Biesty.

**Data curation:** Vivienne Hanrahan.

**Formal analysis:** Vivienne Hanrahan, Linda Biesty.

**Investigation:** Vivienne Hanrahan.

**Methodology:** Vivienne Hanrahan, Linda Biesty.

**Project administration:** Vivienne Hanrahan.

**Supervision:** Katie Gillies, Linda Biesty.

**Writing – original draft:** Vivienne Hanrahan.

**Writing – review & editing:** Vivienne Hanrahan, Katie Gillies, Linda Biesty.

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
