## [Decision Letter · Decision Letter 0]

6 Apr 2020

PONE-D-20-04175

Recruiters' perspectives of recruiting women during pregnancy and childbirth to clinical trials: a qualitative evidence synthesis

PLOS ONE

Dear Dr. Hanrahan,

Thank you for submitting your manuscript to PLOS ONE. After careful consideration, we feel that it has merit but does not fully meet PLOS ONE’s publication criteria as it currently stands. Therefore, we invite you to submit a revised version of the manuscript that addresses the points raised during the review process.

We would appreciate receiving your revised manuscript by May 21 2020 11:59PM. To enhance the reproducibility of your results, we recommend that if applicable you deposit your laboratory protocols in protocols.io, where a protocol can be assigned its own identifier (DOI) such that it can be cited independently in the future. For instructions see: http://journals.plos.org/plosone/s/submission-guidelines#loc-laboratory-protocols

We look forward to receiving your revised manuscript.

Kind regards,

Tim Mathes

Academic Editor

PLOS ONE

Journal Requirements:

1. Please provide any updates you might have since the original search was performed in January 2019, or  provide a specific rationale for ending your search at that time.

2. In the methods, please describe how risk of bias was assessed in individual studies (including specification of whether this was done at the study or outcome level, or both, and the specific test employed, such as the I^2 statistic), and how this information was used in any data synthesis.

3. In the Methods, please specify any assessment of risk of bias that may affect the cumulative evidence (e.g., publication bias, selective reporting within studies). Please ensure that the specific method of assessment (funnel plot, Egger's test, Begg's test, etc) is mentioned.

4. We notice that the email address and names included in the supplementary information file may be potentially identifying. Please ensure that the email is shared only with consent; it may be removed at this time if the sender did not consent to the email being published.

Reviewers' comments:

Reviewer's Responses to Questions

**Comments to the Author**

1. Is the manuscript technically sound, and do the data support the conclusions?

Reviewer #1: Yes

Reviewer #2: Yes

Reviewer #3: Yes

2. Has the statistical analysis been performed appropriately and rigorously? 

Reviewer #1: N/A

Reviewer #2: N/A

Reviewer #3: N/A

3. Have the authors made all data underlying the findings in their manuscript fully available?

Reviewer #1: Yes

Reviewer #2: Yes

Reviewer #3: Yes

4. Is the manuscript presented in an intelligible fashion and written in standard English?

Reviewer #1: Yes

Reviewer #2: Yes

Reviewer #3: Yes

5. Review Comments to the Author

Reviewer #1: Thank you for asking me to peer review this novel qualitative evidence synthesis. It appears to have been well conducted using an appropriate amount of systematic processing. There are however a number of issues that require your attention to further strengthen the manuscript. Some are methodological and some are reporting issues.

1. ENTREQ is a generic reporting guideline for a range of different qualitative evidence synthesis types. The language used in places (such as the abstract) inappropriately elevates this generic QES reporting guideline to the level of a methods manual (which it is not). See for example the abstract 'our synthesis was guided by the ENTREQ statement'. Suggest consider rewriting along the following lines: 'The synthesis was undertaken using Thomas and Harden's three stage thematic synthesis method, and reported following the ENTREQ reporting guideline for generic qualitative evidence syntheses.'

1.1 The manuscript also reports that the review follows PRISMA guidelines. All relevant elements of PRISMA were incorporated into ENTREQ with the caveat that ENTREQ guidelines do not explicitly state that the flow of literature should be reported with a PRISMA type diagram. Please clarify.

2. It is more usual to consider 'the factors that create barriers and enablers to'...

3. The convention as you state is to report number of studies and number of articles. This also needs to be reflected in the flow diagram. The last box needs to reflect number of studies and number of articles.

4. Please cross check reporting against the journal headings and subheadings for reporting reviews and syntheses. See for example your heading 'methodology of synthesis'. This is not a usual heading. 'Synthesis methodology' or just 'Synthesis' is more appropriate. I could not however find a 'data extraction/ abstraction subheading'. As the Thomas and Harden method combines data abstraction and synthesis you could see if a 'Data abstraction and synthesis heading' is agreeable to the journal.

5. PROSPERO is becoming more flexible concerning the protocols they publish. Arguably recruitment rates to clinical trials or not is an important clinical indicator. Did you try registering the review or did you just assume that it did not meet their increasingly flexible criteria?

6. Some of the terminology used to describe included study types is a bit unconventional (such as visual and collaborative studies). Suggest stick to naming the types of types of qualitative methodologies and methods that would meet the inclusion criteria and be consistent throughout the paper.

7. Definition of a finding (p7) 'This review defined a finding as a statement made by a study author'. This feels too restrictive and is open to misinterpretation. A finding can be a theory, a new concept, a typology, a theme, subtheme, findings statement, impact statement etc etc, findings can be presented in tables, diagrams, infograms, pictures, summary statements additional online only files, abstracts, summary bullet points etc etc (ie anywhere in the paper). The suggestion is to rewrite 'what is a finding' and to articulate that findings can be found anywhere in the paper. A theoretical lens for framing findings for example can be found in the introduction or methods section of a primary study.

8. Anthropomorphism. Please avoid giving the review human qualities. The review is an object - it cannot do anything. See for example: 'This review defined...' Rewrite as 'Findings were defined as ........'

9. Data extraction. With the Thomas and Harden's method of thematic synthesis, it is usual to code the entire paper and develop the synthesis using a software product. The text suggests that data were extracted into a table and then coded? It is more usual to extract the study information and characteristics into a table and then upload the entire manuscript and additional files to a software product for coding and synthesis. Please clarify.

10. Synthesis. The analytical themes presented in the findings section appear to be close to the primary studies and do not yet seem to move substantially away to develop new patterns and understandings that were not apparent in single primary studies. This becomes more apparent when you look at the summary of themes and subthemes presented in the supplemental files. The question then arises are these descriptive level themes rather than analytical themes? You may need to be clearer that with 4 studies there appeared to be less opportunity to 'go beyond' the primary studies to analytical level. Given that you only had 4 studies, you may not have been able to advance to a truly analytical level without the input of extensive stakeholder consultation and building in a theory development stage in which you 'bridge the gaps' with new hypotheses. The other option would have been to see if any of the primary qualitative studies were linked to interventions (ie trial sibling studies) where you could have made better use of the data in it's original context to explain how a specific trial recruited and the potential impact of staff behaviours on recruitment rates. The key is not to overplay your findings if they really are not truly 'analytical' level. You may have a combination of descriptive with some reaching up into analytical - but at face value they don't look like stand alone analytical findings that move beyond the primary studies. Simply revising the theme head to something different but similar is not sufficient.

If not able to further advance the findings to a truly analytical level then suggest be transparent and outline the additional strategies that you could potentially have used in the limitations section. Please reconsider, and clarify.

10.1 if able to make a more convincing case for analytical level findings, it would also be helpful for the reader to distinguish between descriptive level and analytical level themes in the text and the tables.

Overall the supplementary online files are very useful. Good to see a table of methodological limitations in each study reported in full.

Reviewer #2: I enjoyed reading this paper. It addressed an important topic, was well written and easy to read. My comments are below:

Abstract conclusion refers to ‘combing’ – should this be ‘combining’?

Methodology- some authors use thematic synthesis in a very broad sense. Suggest in the abstract you make reference to Thomas and Hardy’s approach specifically.

Table 1: please add the reference numbers for included papers otherwise readers need to scan the full reference list to find these.

Reflexivity: I was very pleased to see such information provided. I would suggest though that the information about the researchers is added to your strengths/limitations section on p36 as your professional background is the context in which your interpretations/analysis are made so, it helps readers to understand this when they are reading about the strengths/limitations of your study.

Findings: page 20. A) You note five papers from four studies but this does not match your PRISMA diagram which says five studies. Please amend your PRISMA diagram to say 4 studies, 5 papers. B) Could you report somewhere whether these 4 studies were found via the electronic searching or if any were found via the grey literature search? References suggest these were from the e-searches but, it is useful for readers to know this in terms of study quality. Perhaps add to page 20 all identified papers were found through database searching.

Strengths and limitations and confidence of findings: Perhaps these sections could be merged but with more detail added? You raise key points but you need to go further. For example, your team consists of two midwives and a trial methodologist working in Scotland and Ireland. Team make up e.g. discipline, gender, is likely to have influenced your thematic synthesis. Perhaps a team of obstetricians may have interpreted the original study findings in a slightly different way to you as they would approach this synthesis from a different professional perspective? Please can you add something to this effect in this section of your paper – just the point that your final synthesis reflects your professional and personal backgrounds. Also, please comment that your included studies were from UK and Netherlands so findings reflect a European perspective from two countries with universal healthcare.

Figure 2: I liked this but felt it was incomplete. A) The title needs improved it was too vague e.g. maybe it could mention themes? B) A point for consideration – could the 4 themes/columns be brought together in some way e.g. within a box? Perhaps providing an over-arching theme such as ‘recruiters’ perspectives of recruiting women during pregnancy and childbirth to clinical trials: 4 dimensions informing their practice’. At the moment, it’s simply a visual representation of your descriptive narrative. I feel there is potential for Fig 2 to say more – perhaps linking it to your interpretation/discussion so, it goes beyond just summarising your themes. This sort of thing could give your figure more impact and meaning to readers.

S3 – one column is missing its reference.

Reviewer #3: Thank you for the opportunity to read this Qualitative Evidence Synthesis investigating recruiter experiences and perceptions of barriers and facilitators to recruitment to randomised trials conducted during pregnancy and childbirth.

I enjoyed reading this systematically presented and well written study. Findings include new interpretations, arguing knowledge and power interact and influence recruitment in this context. The authors suggest findings could have relevance to others of the top 10 areas identified for PRIORITY (specifically how best to ensure inclusion of underrepresented and vulnerable groups and how best to ensure manageable workloads for clinical and research staff who recruit to trials - p 35).

There is a lack of research involvement of women recruited during pregnancy and childbirth and this QES will contribute important insights into recruiter perspectives in an area where knowledge is currently lacking.

This work deserves publication. My comments focus mainly on making more explicit 1) the methods of the QES; 2) clarifying findings. Please see the attached document for my detailed comments.

6. PLOS authors have the option to publish the peer review history of their article (what does this mean?). If published, this will include your full peer review and any attached files.

Reviewer #1: Yes: Jane Noyes

Reviewer #2: Yes: Dr Nicola Ann Ring

Reviewer #3: Yes: Julia Wade

---

## [Author Response · Author response to Decision Letter 0]

28 Apr 2020

We would like to thank the reviewers for their time and care in providing us with very helpful feedback. We feel that the manuscript has been greatly enhanced by this contribution.

---

## [Decision Letter · Decision Letter 1]

3 Jun 2020

Recruiters' perspectives of recruiting women during pregnancy and childbirth to clinical trials: a qualitative evidence synthesis

PONE-D-20-04175R1

Dear Dr. Hanrahan,

We’re pleased to inform you that your manuscript has been judged scientifically suitable for publication and will be formally accepted for publication once it meets all outstanding technical requirements.

Kind regards,

Tim Mathes

Academic Editor

PLOS ONE

Additional Editor Comments (optional):

Reviewers' comments:

Reviewer's Responses to Questions

**Comments to the Author**

1. If the authors have adequately addressed your comments raised in a previous round of review and you feel that this manuscript is now acceptable for publication, you may indicate that here to bypass the “Comments to the Author” section, enter your conflict of interest statement in the “Confidential to Editor” section, and submit your "Accept" recommendation.

Reviewer #1: All comments have been addressed

Reviewer #2: All comments have been addressed

Reviewer #3: All comments have been addressed

2. Is the manuscript technically sound, and do the data support the conclusions?

Reviewer #1: Yes

Reviewer #2: Yes

Reviewer #3: Yes

3. Has the statistical analysis been performed appropriately and rigorously? 

Reviewer #1: N/A

Reviewer #2: N/A

Reviewer #3: N/A

4. Have the authors made all data underlying the findings in their manuscript fully available?

Reviewer #1: Yes

Reviewer #2: Yes

Reviewer #3: Yes

5. Is the manuscript presented in an intelligible fashion and written in standard English?

Reviewer #1: Yes

Reviewer #2: Yes

Reviewer #3: Yes

6. Review Comments to the Author

Reviewer #1: (No Response)

Reviewer #2: Thank you for addressing the issues I raised. I enjoyed reading your revised paper and the improvements had strengthened your work.

Reviewer #3: Thank you for the opportunity to review this resubmitted manuscript. I am happy that the issues I raised have been addressed.

I wanted to raise the following 3 minor points for minor edits:

1. P15 (unmarked MS) first paragraph, line 5 – change synthesis to synthesise.

2. P16 first paragraph ‘A summary….WAS…’ and again later same page perhaps: A ‘Summary of Qualitative Findings’ WAS derived from the findings of the thematic synthesis. As the research team summarised and identified the central idea of each finding, the authors then made an overall GRADE-CERQual assessment of confidence based on these findings.

3. P37 Conclusion: suggest the authors bring the reference to ‘knowledge’ in line with Abstract and Findings, both of which now refer to ‘overarching theme of belief and power which links’ (rather than ‘knowledge and power which underpins’). Also correct ‘combing’ to ‘combining’.

7. PLOS authors have the option to publish the peer review history of their article (what does this mean?). If published, this will include your full peer review and any attached files.

Reviewer #1: Yes: Jane Noyes

Reviewer #2: Yes: Dr Nicola A. Ring

Reviewer #3: Yes: Julia Wade

---

## [Editor Report · Acceptance letter]

10 Jun 2020

PONE-D-20-04175R1

Recruiters' perspectives of recruiting women during pregnancy and childbirth to clinical trials: a qualitative evidence synthesis

Dear Dr. Hanrahan:

I'm pleased to inform you that your manuscript has been deemed suitable for publication in PLOS ONE. Congratulations! Your manuscript is now with our production department.

Kind regards, 

on behalf of

Dr. Tim Mathes 

Academic Editor

PLOS ONE